# Biological evaluation of a mechanical ventilator that operates by controlling an automated manual resuscitator. A descriptive study in swine

**Maryanne Melanie Gonzales Carazas[1], Cesar Miguel Gavidia[2], Roberto Davila Fernandez[2], Juan Alberto Vargas Zuñiga[3], Alberto Crespo Paiva[4], William Bocanegra[2], Joan Calderon[2], Evelyn Sanchez[2], Rosa Perales[2], Brandon Zeña[4], Juan Fernando Calcina Isique[2], Jaime Reategui[5], Benjamin Castañeda[1,6], Fanny L. Casado[1,6]***

1 Institute of Omics and Applied Biotechnology, Pontificia Universidad Catolica del Peru, Lima, Peru,
2 School of Veterinary Medicine, Universidad Nacional Mayor de San Marcos, Lima, Peru, 3 Centro de Urgencias Veterinarias (CUVET), Lima, Peru, 4 Instituto Veterinario de Oftalmologia (IVO), Lima, Peru,
5 Brein, Lima, Peru, 6 Department of Engineering, Pontificia Universidad Catolica del Peru, Lima, Peru

* fanny.casado@pucp.edu.pe

**Data Availability Statement:** All relevant data are within the paper and its Supporting Information files.

## Abstract

The Covid-19 outbreak challenged health systems around the world to design and implement cost-effective devices produced locally to meet the increased demand of mechanical ventilators worldwide. This study evaluates the physiological responses of healthy swine maintained under volume- or pressure-controlled mechanical ventilation by a mechanical ventilator implemented to bring life-support by automating a resuscitation bag and closely controlling ventilatory parameters. Physiological parameters were monitored in eight sedated animals ($t_0$) prior to inducing deep anaesthesia, and during the next six hours of mechanical ventilation ($t_{1-7}$). Hemodynamic conditions were monitored periodically using a portable gas analyser machine (i.e. BEecf, carbonate, $SaO_2$, lactate, pH, $PaO_2$, $PaCO_2$) and a capnometer (i.e. $ETCO_2$). Electrocardiogram, echocardiography and lung ultrasonography were performed to detect *in vivo* alterations in these vital organs and pathological findings from necropsy were reported. The mechanical ventilator properly controlled physiological levels of blood biochemistry such as oxygenation parameters ($PaO_2$, $PaCO_2$, $SaO_2$, $ETCO_2$), acid-base equilibrium (pH, carbonate, BEecf), and perfusion of tissues (lactate levels). In addition, histopathological analysis showed no evidence of acute tissue damage in lung, heart, liver, kidney, or brain. All animals were able to breathe spontaneously after undergoing mechanical ventilation. These preclinical data, supports the biological safety of the medical device to move forward to further evaluation in clinical studies.

**Funding:** FLC received funding public Peruvian funding from the grant # 055-2020-FONDECYT administered by the Fondo Nacional de Desarrollo Científico, Tecnológico y de Innovación Tecnológica. The funders had no role in study design, data collection and analysis, decision to publish, or preparation of the manuscript.

**Competing interests:** The authors have declared that no competing interests exist.

## Introduction

The most common clinical complication identified in COVID-19 is respiratory distress (29%) with 32% of this requiring ICU care [1]. The main cause of death in Peru is acute respiratory failure quantified by low oxygen saturation and low values of the relationship between arterial pressure $O_2$ and inspired fraction of $O_2$ [2]. The hazard ratio for the estimated the risk of death in COVID-19 patients increased from 1.93–4.71 in patients with oxygen saturation lower than 90% up to 9.13 times at 80% [2]. Countries like Peru, which before the pandemic had a limited hospital infrastructure and lacking sufficient equipment, have seen their health system overloaded and overwhelmed. Despite a 113% increased of ICU beds capacity, just in Lima there was 87% occupancy in the periods from September to December 2020 (between first and second wave) versus 100% occupancy between January and May 2021 (second wave) [3]. To alleviate this need, a multidisciplinary team of professionals designed and developed Masi (meaning friend or partner in the Quechua language), a mechanical ventilator for the COVID-19 emergency as the first open source mechanical ventilator to be mass-produced in Peru [4].

The mechanisms of action traditionally used in mechanical ventilators include: (a) opening valves proportionally to the desired flow of oxygen from a gas system, (b) operating pistons to mobilize gas to the patient, or (c) using turbines to mix filtered ambient air with oxygen to obtain the required fraction of inspired oxygen. Masi uses a novel mechanism consisting of filling a self-inflating resuscitation bag (Fig 1) with oxygen and controlling its mixing with ambient air and mobilization to the patient by compression of the bag. This mechanism uses less oxygen and its fabrication costs are significantly lower than established mechanical ventilators used in ICU.

Masi is an automated ventilator that works in three different operating modes based on the mandatory ventilation controlled by either volume (PC-CMV) or pressure (VC-CMV), and support ventilation under positive pressure (PSV). Additionally, the device senses and regulates inspiration and expiration parameters, respiratory rate, and oxygen flow. Masi was designed to meet the requirements established by the Pan American Health Organization/ World Health Organization (PAHO/WHO) [5] and the Medicines and Healthcare products Regulatory Agency (MHRA, United Kingdom) [6] for mechanical ventilators to manage COVID-19 atypical pneumonia in health services. Performance and electrical safety of Masi were validated using international standards and approved technical tests at the engineering laboratory level [4] using a test lung simulator in a metrology laboratory using the setup shown in Fig 1.

Respiratory translational studies use swine models to study mechanical ventilation due to their anatomical and histological similarities to human clinical endpoints. Therefore, this study uses pigs as a relevant model for the preclinical study of a novel type of mechanical ventilator such as Masi by assessing respiratory physiological parameters such as gas exchange and pulmonary function, acid-base disturbances, lactate and carbonate concentrations, cardiac output and heart rate before, during, and after invasive intubation.

## Materials and methods

### Experimental design

This study aims to determine the safety and efficacy of acute exposure to mechanical ventilation with Masi in healthy swine (Fig 2). The experimental design is in line with the Animal welfare requirements section (part 2) from ISO 10993 –"Biological evaluation of medical devices" to minimize the number of animals required and any pain or distress. A minimum

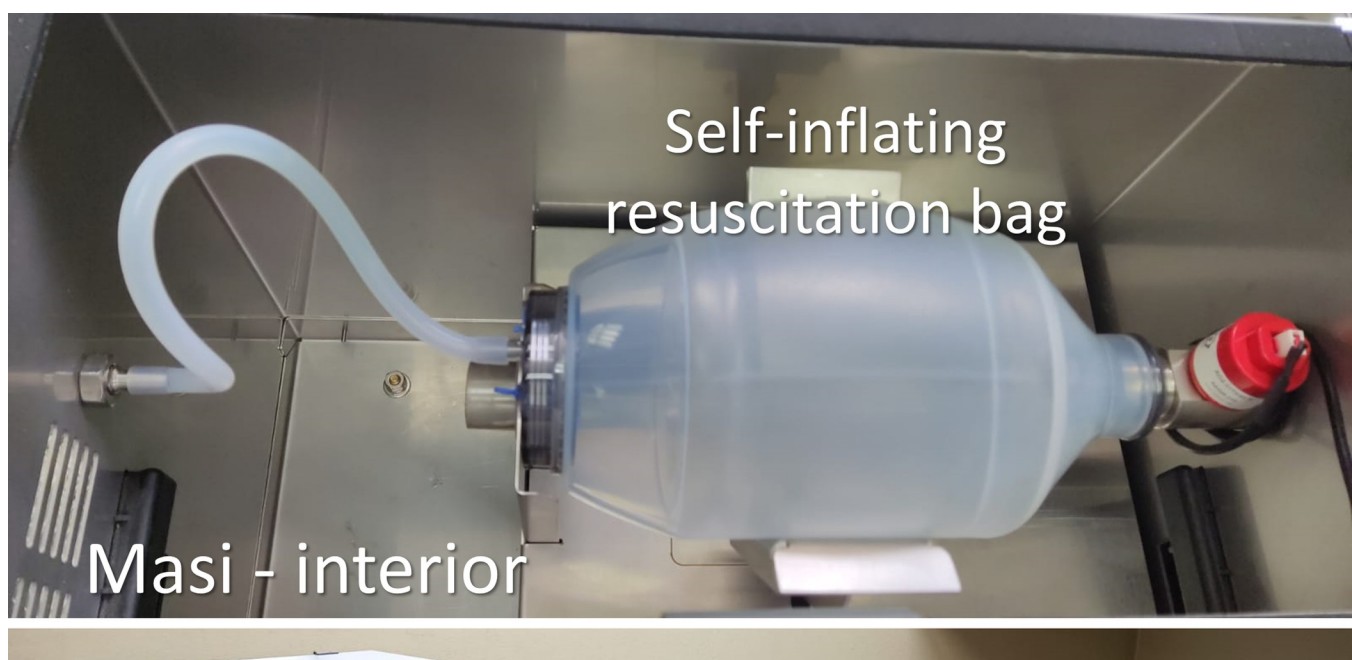

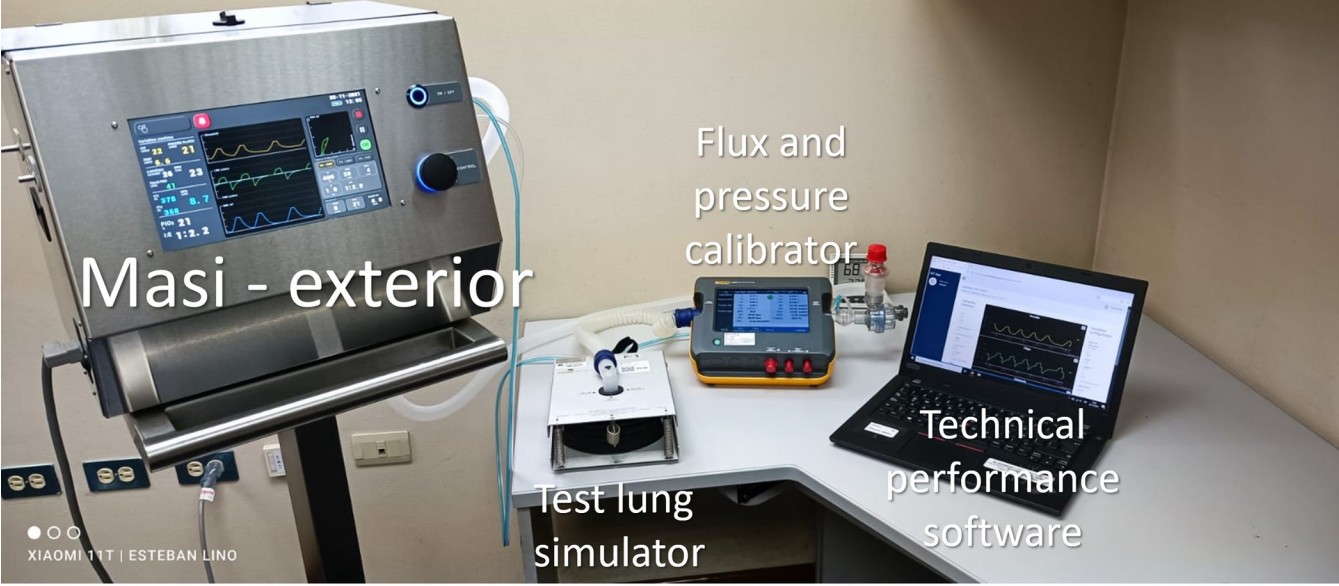

**Fig 1. Experimental setup for the metrological evaluation of Masi.** The self-inflating resuscitation bag was connected in the interior of Masi and the metrological evaluation of the performance of the control of oxygen flow and pressure was done using a test lung simulator, a flux and pressure calibrator and software to validate the response and repeatability of the measurements.

expectation of 50% survival was the criterion to continue the study after the preliminary study and the first group of four pigs of the pre-clinical study.

The preliminary study subjected two pigs to mechanical ventilation for one hour to establish the protocol of the intervention, determine the range of responses of the swine and establish the safety of the ventilator for one hour based on post-mortem evidence to proceed with longer exposure for 6 hours.

A longitudinal pre-clinical study assessed the performance of the device by analysing biological parameters during mechanical ventilation of eight anesthetized pigs for 6 hours. Partial

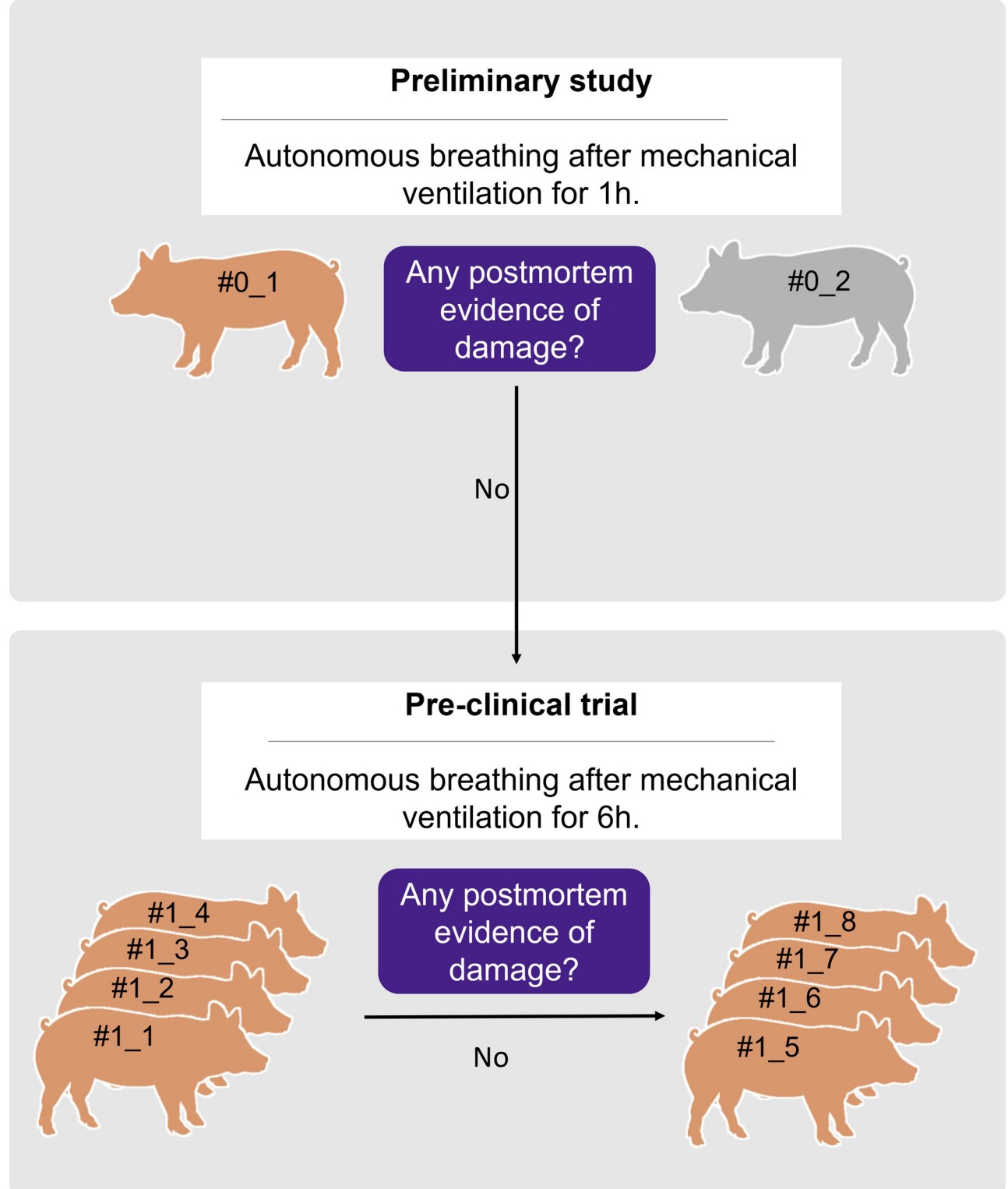

**Fig 2. Experimental design.** A preliminary study optimized anaesthesia dosing and evaluated autonomous breathing after mechanical ventilation for one hour. Since no evidence of gross pathology damage was found, the pre-clinical trial was performed. The trial included a total of eight animals whose autonomous breathing was evaluated after mechanical ventilation for six hours. During the trial, the results from the first group of four individuals were done and since no post-mortem evidence of damage to internal organs was found, the responses from a second group of four individuals were further studied.

results were analysed after four swine to confirm the safety of the ventilator at 6 hours based on physiological responses during the intervention and post-mortem evidence. Next, four more animals were intervened to complete assessing the performance of the device. The statistical validity of the sample assumed a 95% confidence level, 5% margin of error, and that 99.5% of swine are expected to be able to breathe autonomously after six hours of mechanical ventilation with Masi.

## Participants

All pre-clinical protocols carried out in this project were performed in accordance with the Animal Research: Reporting In Vivo Experiments (ARRIVE) guidelines and were approved by the Committee on Research Ethics for Life Sciences and Technologies at Pontificia Universidad Catolica del Peru (Protocol N˚002-2020-CEICVyTech/PUCP). The study used young animals from a modern and commercial pig farm (Universidad Nacional Agraria La Molina, Lima, Peru). For this purpose, we collected standard physiological parameters for different studies involving swine as summarized in Table 1.

A total of eight animals (7 males and 1 female), Yorkshire x Landrace swine, three to four months old weighing 40–60 kg were used in this study as described in Table 2.

The pigs were housed in the animal facilities of the School of Veterinary Medicine at the Universidad Nacional Mayor de San Marcos (UNMSM, Lima, Peru). These animals were kept in the facilities for 3–4 days to allow acclimatization and clinical observation by Veterinarians. The animals received commercial food twice a day (approximately 2 kg/day), and water ad libitum. Pigs were apparently healthy; no clinical signs were seen or reported by the time of the trial. Haematological and biochemical test, as well as ultrasound images from heart and lungs, were performed. It was considered at that time that all of the animals were in good health to participate in biological evaluations for the Masi ventilator.

Accordingly, Fig 3 summarizes the experimental timeline of each intervention, considering sedation and deep anaesthesia protocols, as well as the interventional procedures for sampling and data collection. These procedures will be described in detail below.

**Table 1. Summary of standardized physiological values on cardiorespiratory parameters in swine.**

| Parameters | Mean | SD | n | References |
|---|---|---|---|---|
| Rectal Temperature (˚C) | 38.9 | 0.6 | 30 | [7] |
| pH, in arterial blood | 7.414 | 0.054 | 86 | [7–10] |
| $HCO_3^-$ (mmHg) in arterial blood | 23.7 | 7.7 | 38 | [7, 10] |
| Lactate (mmol/l) in arterial blood | 1.46 | 0.46 | 25 | [9] |
| BEecf (mmol/l) in arterial blood | 0.06 | 6.60 | 38 | [7, 10] |
| $SaO_2$ (%) | 98.2 | 1.67 | 56 | [8–10] |
| $PaO_2$ (mmHg) | 161.9 | 36.4 | 86 | [7–10] |
| $PaCO_2$ (mmHg) | 39.5 | 8.9 | 86 | [7–10] |
| Respiratory rate (breaths/min) | 31 | 8.8 | 30 | [7] |
| Heart rate (beats/min) | 114 | 14.1 | 53 | [7, 8] |
| Cardiac output (l/min) | 5.542 | 1.114 | 23 | [8] |
| Haemoglobin (g/dl) | 8.5 | 1.74 | 25 | [9] |
| Haematocrit (%) | 25.0 | 5.1 | 25 | [9] |

Data was reported as mean ± standard deviation adapted from data available in the literature. SD: standard deviation, pH: potential of hydrogen, $HCO_3^-$: bicarbonate ion, BEecf: base excess in the extracellular fluid compartment, $SaO_2$: oxygen saturation in arterial blood, $PaO_2$: oxygen pressure in arterial blood, $PaCO_2$: carbon dioxide pressure in arterial blood, $ETCO_2$: end-tidal carbon dioxide concentration.

**Table 2. General animal health data from animals participating in the pre-clinical trial.**

| Swine # | Weight (kg) | Age (months) | Sex | Media compliance (ml/cmH$_2$O) | Haemoglobin (g/dl) | Haematocrit (%) |
|---|---|---|---|---|---|---|
| 1 | 41.0 | 3.0 | Male | 39 | 10.2 | 31.5 |
| 2 | 45.5 | 3.0 | Male | 44 | 9.4 | 28.7 |
| 3 | 50.0 | 3.5 | Male | 32 | 9.1 | 26.9 |
| 4 | 49.0 | 3.5 | Male | 39 | 10.5 | 31.3 |
| 5 | 46.5 | 3.5 | Female | 39 | 11.1 | 33.2 |
| 6 | 52.0 | 3.5 | Male | 38 | 10.0 | 31.6 |
| 7 | 56.0 | 4.0 | Male | 39 | 10.3 | 33.1 |
| 8 | 50.0 | 4.0 | Male | 37 | 10.1 | 32.3 |

## Premedication

Premedication with sedative drugs and anaesthesia protocols were performed as previously described by Clauss and co-workers [11]. Animals were food-deprived for 12 hours before the procedures to avoid both gastric dilatation and vomiting. The access to water was also restricted 2–6 hours prior to the process begins. On the morning intervention, swine were sedated with ketamine 20 mg/kg + azaperone 1 mg/kg via intramuscular in the neck, behind the ear. Later, the animals were placed on the work table and receive about 5 minutes of oxygen flow at 6–8 l/min through a mask. The marginal ear vein was catheterized for the administration of both fluids and drugs. The saline solution 0.9% (p/v) was administered at 5 ml/kg/h, and also depending of pig clinical evaluation. Ophthalmic ointment was applied to prevent corneal drying.

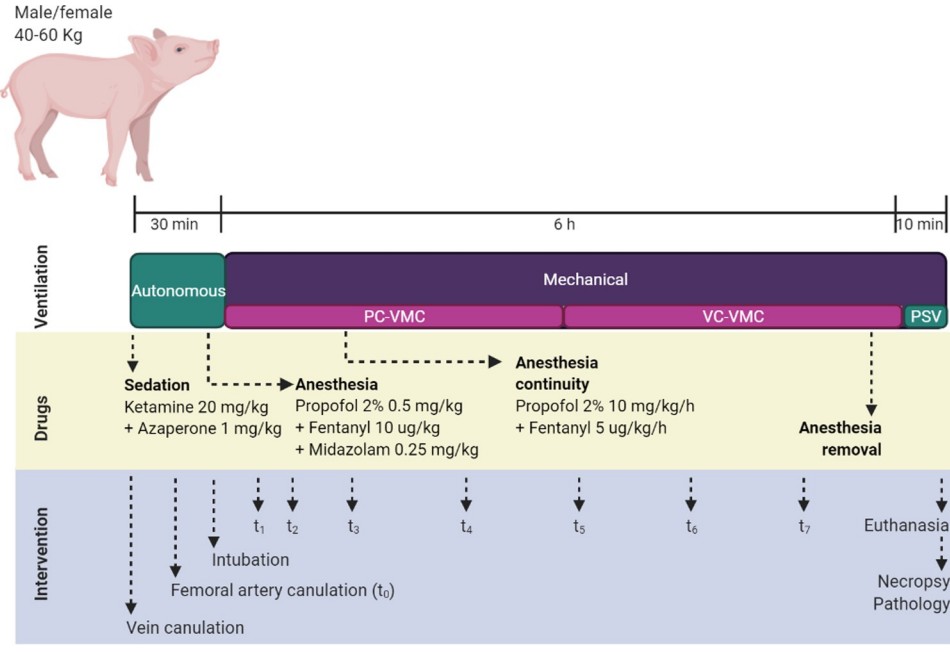

**Fig 3. Timeline for experimental procedures.** The diagram shows the time-points when drugs were supplied, ventilation was controlled and interventions were performed. PC-CMV: Pressure Control-Continuous Mandatory Ventilation, VC-CMV: Volume Control-Continuous Mandatory Ventilation, PSV: Pressure support ventilation, $t_0$ to $t_7$: blood collection time-points.

## Anaesthesia induction

Propofol 0.5–1 mg/kg combined with midazolam 0.25 mg/kg were administered to induce anaesthesia in the swine through a catheterized ear vein. Anaesthetics plane was maintained by a combination of propofol 2.5–5 mg/kg/h during the intervention, in addition to fentanyl 2.5 μg/kg/h if required. After anesthetizing the animal, the femoral artery was dissected to allow taking serial blood samples; this catheter was maintained with sodium heparin solution 5000 IU/ml. Then, animals were placed in ventral decubitus and intubated with a 6–7 mm endotracheal tube, according to animal weight.

Deep anaesthesia was monitored by Veterinarians and maintained in the proper range. While inducing anaesthesia, heart and respiratory rates were measured with a stethoscope. Electrocardiograms (EKG) were performed to monitor cardiovascular responses during the intervention. In addition to vital signs, anaesthetic depth was constantly assessed by jaw tonicity and reflex (corneal touch, pedal flexion, coronary band pinch).

After six hours of mandatory mechanical ventilation with Masi, deep anaesthesia was suspended and swine were switched to a non-mandatory ventilation mode (pressure-support ventilation, PSV). The ability of animals to breath autonomously were monitored until three complete breaths were performed using an apnoea time of 1 minute before proceeding with the euthanasia.

## Lung mechanics analysis

After deep anaesthesia was reached, animals were allowed 10 minutes to stabilize. Then, endotracheal intubation was performed in swine to connect the mechanical ventilator. An experienced veterinary anaesthesiologist monitored the ventilatory parameters by capnography and capnometry. Ventilator parameters were calculated to obtain tidal volumes of about 6 ml/kg; and set positive end expiratory pressure (PEEP) between 7–9 cm $H_2O$, and an inspiration-expiration ratio (I:E) about 1:2 with tidal volumes between 450 to 550 ml. According to literature, our protocol provided a protective ventilation once it considered low tidal volume, and high PEEP [12]. Once the parameters were stabilized, the constant pressure-controlled ventilator mode (PC-CMV) was used for the firsts three hours and then the operating mode was replaced for a constant volume control (VC-CMV) for additional three hours.

## Assessment of physiological parameters

Complete haemogram and serum biochemistry analysis were performed to determine their physiological conditions prior to the intervention. The ProCyte Dx haematological analyser (Idexx Laboratories) and Catalyst One (Idexx Laboratories) were utilized to achieve this purpose, respectively.

Oxygen saturation, blood pressure, respiratory and heart rate were constantly monitored with a veterinary multi-parametric monitor through a cuff placed in the hind leg. Temperature was also registered with a rectal probe and maintained above 36°C during sedation using infrared heating lamps when necessary.

Femoral artery catheterization was performed managing lidocaine HCl 2% locally; arterial samples were collected in a 1 ml heparinized syringe and immediately analysed using the iSTAT system CG4+ cartridge (Abbott Point of Care Inc). Base excess in the extracellular fluid compartment (BEecf), bicarbonate ($HCO_3^-$), arterial oxygen saturation ($SaO_2$), lactate, pH, arterial pressure of oxygen ($PaO_2$), and arterial pressure of carbon dioxide ($PaCO_2$) were measure in arterial blood by gas analyser. Heart and respiratory rates were determined by auscultation with a stethoscope. End-tidal carbon dioxide ($ETCO_2$) were measured with a capnograph Respironics LoFlo Side-Stream $CO_2$ Sensor Module (©Philips). Arrhythmia's presence was

monitored using the computerized electrocardiogram EG PC (TEB®). Echocardiography and lung ultrasonography were performed with the scanner MyLab$^{TM}$30 Vet Gold (Esaote S.p.A.) ultrasound machine.

Samples and data were collected in specific ranks of time. Prior to the anaesthesia, a baseline ($t_0$) sample was considered for each swine. Once mechanical ventilation with Masi starts, seven additional times ($t_{1-7}$) to collect samples and data were considered. The $t_{1-7}$ period was carried out according to the following pattern: 15 min ($t_1$), 30 min ($t_2$), 60 min ($t_3$), 120 min ($t_4$), 180 min ($t_5$), 240 min ($t_6$), and 300 min ($t_7$).

### Euthanasia, necropsy and histopathology

Pharmacological euthanasia was performed in accordance with the ethical regulation for the humane treatment of experimental animals. Hence, animals received an overdose of sodium pentobarbital (150 mg/kg) intravenously [13]. Animal death was confirmed by auscultation of heart and respiratory rate with a stethoscope. Necropsy was performed following standard procedures for pigs [14]. Macroscopic signs of lesions in the brain, liver, kidney, heart, and lungs were evaluated. Furthermore, samples were taken for histopathological studies of the brain, heart, kidney, liver, and lungs to look for evidence of barotrauma or acute hypoxia.

### Statistical analysis

Statistic 10.0 software (StatSoft Europe, Hamburg 22301, Germany) was used for the statistical analysis. For descriptive purposes, mean, median, standard error of the mean, first and third quartile, minimum and maximum values were calculated from different sampling points. The absence of normal distribution was determined by Shapiro Wilk's W test. Grubbs's test determined significant outliers in our data (GraphPad QuickCalcs, www.graphpad.com). In addition, the Friedman non-parametric test for repeated measures compared ranks over sampling points. The Wilcoxon Matched Pairs test compared between intervals and assessed median variations by comparing adjacent time-periods, and contrasting each experimental period with the baseline. Non-parametric Spearman rank-order correlation analysis assessed the relationship between quantitative variables. GraphPad Prism 6.0 (GraphPad Software, California 92108, USA) plotted our data. Baseline parameters were compared with literature data using Student T-test, assuming that the data from all of the other studies and ours had a normal distribution. For all the assays, statistical significance was set at p-value $\leq$ 0.05.

## Results and discussion

Haematological analysis performed in swine during housing period exhibited that physiological measurements were within standard normal limits. Nevertheless, pre-intervention echocardiograms showed that two of the patients (pig 2 and pig 5) presented sub-clinical heart valve pathology. Later, histopathological analyses determined a chronic inflammation of the lungs that might be attributed to an enzootic pneumonia of high prevalence in Peru caused by *Mycoplasma hyopneumoniae* (S1 Table and S1B Fig in S1 File). However, these chronic pathologies would not interfere with the observation of barotrauma acute lung lesions that are of interest in our experimental design.

The average for swine's cardiorespiratory and blood biochemistry parameters taken at the initial sampling procedure ($t_0$) are displayed in Table 3. No significant differences were found in most of the studied parameters when comparing previously published data for swine under sedation (Table 1), with the exception of $HCO_3^-$, BEecf, $SaO_2$, and $PaO_2$ levels (S2 Table in S1 File).

Table 3. Summary of baseline values ($t_0$) for the swine participating in the pre-clinical trial.

| Parameters | n | Mean | Min. | $Q_1$ | Median | $Q_3$ | Max. | SEM |
|---|---|---|---|---|---|---|---|---|
| Temperature (˚C) | 8 | 36.6 | 35.9 | 36.3 | 36.7 | 36.8 | 37 | 0.1 |
| pH | 8 | 7.395 | 7.329 | 7.368 | 7.410 | 7.422 | 7.431 | 0.013 |
| $HCO_3^-$ (mmHg) | 8 | 32.3 | 29.8 | 30.6 | 32.3 | 33.8 | 34.8 | 0.6 |
| Lactate (mmol/l) | 8 | 1.41 | 0.63 | 1.23 | 1.36 | 1.80 | 1.88 | 0.15 |
| BEecf (mmol/l) | 8 | 7.2 | 5 | 5.5 | 7 | 9 | 10 | 0.7 |
| $SaO_2$ (%) | 8 | 100 | 100 | 100 | 100 | 100 | 100 | 0.0 |
| $PaO_2$ (mmHg) | 8 | 334 | 169 | 278 | 322 | 387.5 | 529 | 37.6 |
| $PaCO_2$ (mmHg) | 8 | 52.9 | 45.1 | 48.3 | 52.6 | 57.6 | 61.4 | 2.0 |
| $ETCO_2$ (mmHg) | 8 | 33.7 | 31 | 32 | 34 | 35.5 | 36 | 0.7 |
| Respiratory rate (breaths/min) | 8 | 37.1 | 20 | 28.5 | 38 | 44 | 56 | 4.1 |
| Heart rate (beats/min) | 8 | 85.4 | 69 | 77.5 | 83.5 | 93.5 | 105 | 4.3 |
| Cardiac output (l/min) | 8 | 5.892 | 3.203 | 4.752 | 6.087 | 7.040 | 8.171 | 0.586 |

After sedation but prior to the induction of deep anesthesia required for connection with Masi mechanical ventilator, cardiorespiratory and physiological parameters were collected for the animals to be used as baseline values. Min.: minimum value, Max.: maximum value, $Q_1$: lower quartile, $Q_3$: upper quartile, SEM: standard error of the mean, pH: potential of hydrogen, $HCO_3^-$: bicarbonate ion, BEecf: base excess in the extracellular fluid compartment, $SaO_2$: arterial oxygen saturation, $PaO_2$: arterial oxygen pressure, $PaCO_2$: arterial carbon dioxide pressure, $ETCO_2$: end-tidal carbon dioxide.

Fig 4 shows the trends of the parameters over time. Atypical responses in Pig 2 and Pig 5 were observed for bicarbonate, BEecf, lactate, and $ETCO_2$, that might be explained by underlying sub-clinical heart valve pathologies.

According to Wilcoxon test, perceived in Fig 4, the levels of $SaO_2$, $PaO_2$, and BEecf dropped drastically when assessed at $t_1$, with respect to $t_0$. In addition, variations in pH and $PaCO_2$ values were observed when contrasting $t_1$ and $t_2$. Although some of the values dropped from $t_0$ to $t_1$, most of them increased or returned to initial values, in the following measures. Based on these differences, $t_1$ was considered as a temporary period of transition from sedation to deep anaesthesia, with limited relevance on the effect of ventilation. Therefore, $t_1$ values were not considered when summarizing the physiological effects attributed to the use of the Masi mechanical ventilator shown in Table 4.

Descriptive statistics for the physiological parameters analysed during the use of the Masi mechanical ventilator ($t_{2-7}$) are described in Table 4. Complete data is provided in S3 Table in S1 File. The rank comparison performed through Friedman's non-parametric test for repeated measures evidenced significant differences in $HCO_3^-$, lactate, BEecf, $PaO_2$, and $ETCO_2$ levels over time. To identify the variability points, the Wilcoxon test by pairs was performed. Among the sampling points $t_{2-7}$, both $HCO_3^-$ and BEecf values gradually increased over time, lactate concentrations decrease, meanwhile $PaO_2$ and $PaCO_2$ remained fluctuating (Fig 5). Since comparisons were performed using median values, outlier data should not interfere with the analysis.

Data from physiological parameters documented during exposure to Masi mechanical ventilation ($t_{2-7}$) were contrasted, one by one, with baseline values ($t_0$). Table 5 shows us the dimension of the differences found through the z-score, and the statistical significance supported by the Friedman test.

According to Table 5, strong variations of some parameters stabilized in time until reaching back to baseline values. For instance, the $HCO_3^-$, which initially ($t_1$) suffered a drop in its concentration levels (Fig 4), gradually increased its concentration in arterial gases through $t_{2-7}$ period (Fig 5). These progressive changes allowed reducing the differences with baseline levels

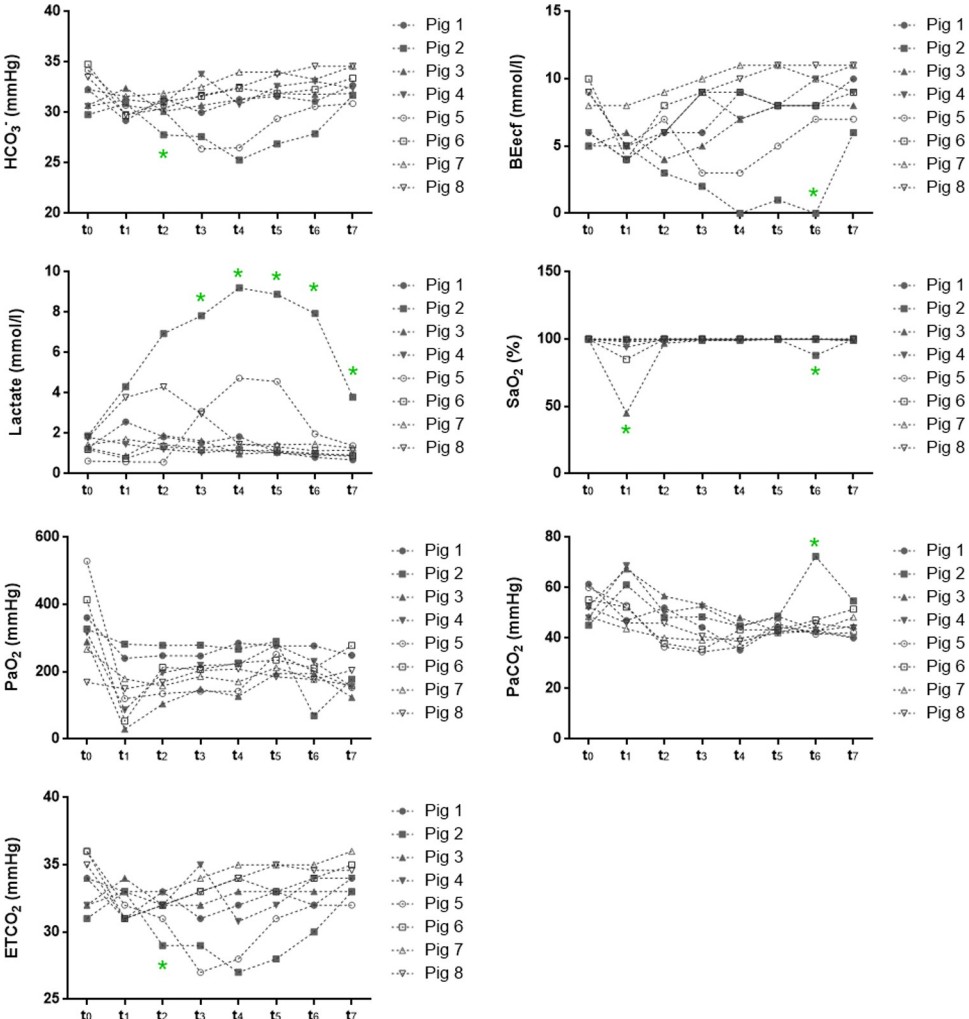

**Fig 4. Responses over time of arterial blood biochemical markers to mechanical ventilation.** Blood samples from each pig were collected at specific time-points and blood biochemistry was assessed using a point-of-care device. pH: potential of hydrogen, $HCO_3^-$: bicarbonate ion, BEecf: base excess in the extracellular fluid compartment, $SaO_2$: arterial oxygen saturation, $PaO_2$: arterial oxygen pressure, $PaCO_2$: arterial carbon dioxide pressure, $ETCO_2$: end-tidal carbon dioxide, *: outlier measurements.

($t_0$). Similarly, $ETCO_2$ presented significant differences in contrast to baseline initially ($t_2$). Through its evolution over time, the differences in median values between experimental data and baseline were reduced. In addition, BEecf and lactate values underwent changes during the initial phase ($t_1$) that were recovered by $t_5$. It is worth noting that changes in pH and $PaO_2$ altered in the initial period with the mechanical ventilator never reached back to baseline levels. The experimental median values of pH ($t_{2-7}$) showed significant differences from the baseline median ($t_0$), excepting at $t_2$ and $t_6$. In the evolutionary graph for this parameter (Fig 4) it was observed that, despite the initial reduction in pH ($t_1$), experimental values ($t_{2-7}$) tend to be higher than the basal ones. The $PaO_2$ showed significant differences from baseline throughout the experimental time, suffering a dramatic fall at the beginning of the experimental protocol ($t_1$) and it never recovered (Fig 4). However, the variance between the experimental values ($t_{2-7}$, exposure to Masi) and the baseline ($t_0$) remains constant over time, neglecting the effect of fluctuations registered for this period ($t_{2-7}$, Fig 5). While $SaO_2$ did not show significant

**Table 4. Summary of statistical distribution for arterial blood biochemistry values ($t_{2-7}$).**

| Parameters | Mean | SEM | Min. | $Q_1$ | Median | $Q_3$ | Max. | n | Friedman test p-value |
|---|---|---|---|---|---|---|---|---|---|
| pH | 7.455 | 0.010 | 7.193 | 7.418 | 7.473 | 7.506 | 7.557 | 48 | 0.744 |
| $HCO_3^-$ (mmHg) | 31.3 | 0.3 | 25.3 | 30.5 | 31.6 | 32.5 | 34.6 | 48 | 0.003 |
| Lactate (mmol/l) | 2.29 | 0.33 | 0.57 | 1.05 | 1.34 | 1.94 | 9.21 | 48 | 0.003 |
| BEecf (mmol/l) | 7.3 | 0.4 | 0 | 6 | 8 | 9 | 11 | 48 | 0.021 |
| $SaO_2$ (%) | 99.5 | 0.3 | 88 | 100 | 100 | 100 | 100 | 48 | 0.335 |
| $PaO_2$ (mmHg) | 203 | 7.7 | 69 | 164 | 204 | 247.5 | 290 | 48 | 0.014 |
| $PaCO_2$ (mmHg) | 44.8 | 0.9 | 34.6 | 40.5 | 43.6 | 48.2 | 72.4 | 48 | 0.235 |
| $ETCO_2$ | 32.5 | 0.3 | 27 | 32 | 33 | 34 | 36 | 48 | 0.002 |

Descriptive statistics like mean, standard error of the mean (SEM), minimum (Min.), 25th percentile ($Q_1$), median, 75th percentile ($Q_3$), maximum (Max.), and number of observations (n) for each cardiopulmonary parameter measure over the six hours of mechanical ventilation with Masi ($t_{2-7}$) were summarized below. Significant differences when using the Friedman test to analyse differences among $t_{2-7}$ were established at p-value <0.05. pH: potential of hydrogen, $HCO_3^-$: bicarbonate ion, BEecf: base excess in the extracellular fluid compartment, $SaO_2$: arterial oxygen saturation, $PaO_2$: arterial oxygen pressure, $PaCO_2$: arterial carbon dioxide pressure, $ETCO_2$: end-tidal carbon dioxide.

differences at any time ($t_{2-7}$) when compared with basal values, fluctuations observed in $PaCO_2$ ($t_{2-7}$, Fig 5) produced significant variations between experimental and basal medians only at specific times, $t_4$ and $t_5$.

A Spearman correlation analysis between $ETCO_2$ and $PaCO_2$ was performed to determine possible effects on lung function that might by carried to blood biochemistry quality [15, 16]. Spearman test indicates that there is no significant correlation between both variables (Fig 6).

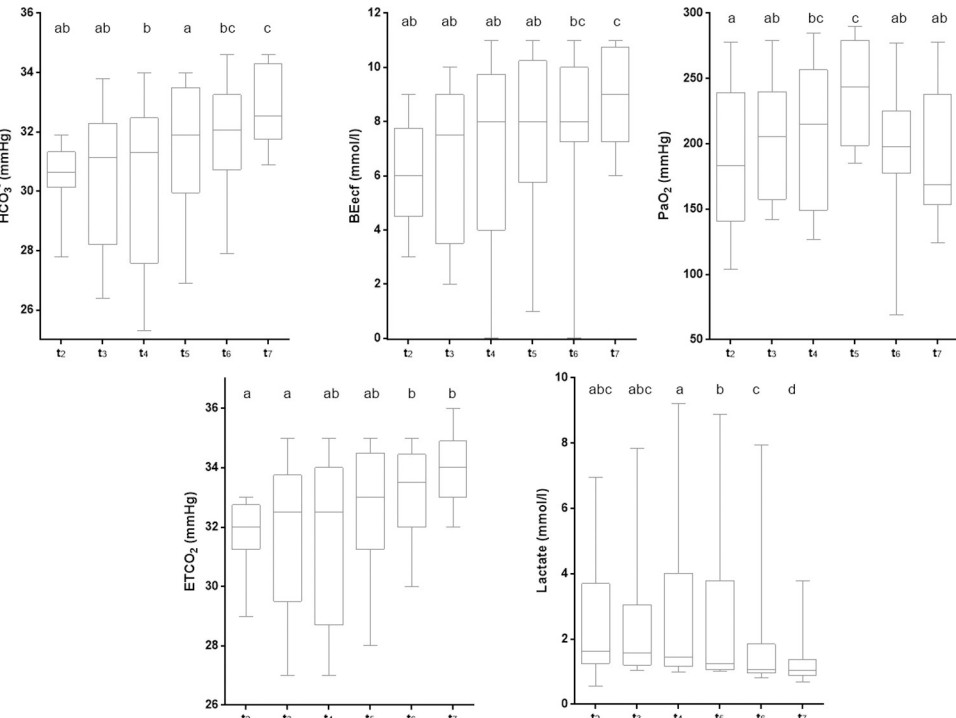

**Fig 5. Distribution of physiological variables over time.** Median, lower and upper quartiles are illustrated for bicarbonate, lactate, base excess, oxygen, and carbon dioxide arterial pressure. Significant differences among time-points are represented by letters on top of the box-plots. Distributions at time-points with different letters are significantly different. $HCO_3^-$: bicarbonate ion, BEecf: base excess in the extracellular fluid compartment, $PaO_2$: arterial oxygen pressure, $PaCO_2$: arterial carbon dioxide pressure, $ETCO_2$: end-tidal carbon dioxide.

**Table 5. Changes in arterial blood biochemical parameters per time point adjusted to baseline.**

| Parameters | Time points | | | | | |
|---|---|---|---|---|---|---|
| | $t_2$ | $t_3$ | $t_4$ | $t_5$ | $t_6$ | $t_7$ |
| pH | 1.330 | 1.960 | 2.100 | 1.960 | 1.400 | 2.100 |
| | (0.183) | (0.049) | (0.036) | (0.049) | (0.161) | (0.036) |
| $HCO_3^-$ (mmHg) | 2.380 | 1.268 | 1.400 | 0.700 | 0.770 | 0.840 |
| | (0.017) | (0.205) | (0.161) | (0.484) | (0.441) | (0.401) |
| Lactate (mmol/l) | 1.521 | 1.400 | 0.420 | 0.169 | 0.280 | 0.560 |
| | (0.128) | (0.161) | (0.674) | (0.866) | (0.779) | (0.575) |
| BEecf (mmol/l) | 1.887 | 0.539 | 0.140 | 0.070 | 0.560 | 1.820 |
| | (0.059) | (0.589) | (0.889) | (0.944) | (0.575) | (0.069) |
| $SaO_2$ (%) | 1.342 | 1.342 | 1.342 | - | - | 1.603 |
| | (0.179) | (0.179) | (0.179) | | | (0.109) |
| $PaO_2$ (mmHg) | 2.366 | 2.380 | 2.380 | 2.380 | 2.380 | 2.380 |
| | (0.018) | (0.017) | (0.012) | (0.017) | (0.017) | (0.017) |
| $PaCO_2$ (mmHg) | 1.680 | 1.680 | 2.520 | 2.100 | 1.400 | 1.680 |
| | (0.093) | (0.093) | (0.017) | (0.036) | (0.161) | (0.093) |
| $ETCO_2$ (mmHg) | 2.201 | 1.363 | 1.960 | 1.363 | 0.840 | 0.592 |
| | (0.028) | (0.179) | (0.049) | (0.173) | (0.401) | (0.554) |

Statistical differences between the medians of the data obtained at each sampling point ($t_{2-7}$) and the median of the baseline data ($t_0$) were analysed using the Friedman test. The table shows the z-score (p-value). pH: potential of hydrogen, $HCO_3^-$: bicarbonate ion, BEecf: base excess in the extracellular fluid compartment, $SaO_2$: arterial oxygen saturation, $PaO_2$: arterial oxygen pressure, $PaCO_2$: arterial carbon dioxide pressure, $ETCO_2$: end-tidal carbon dioxide.

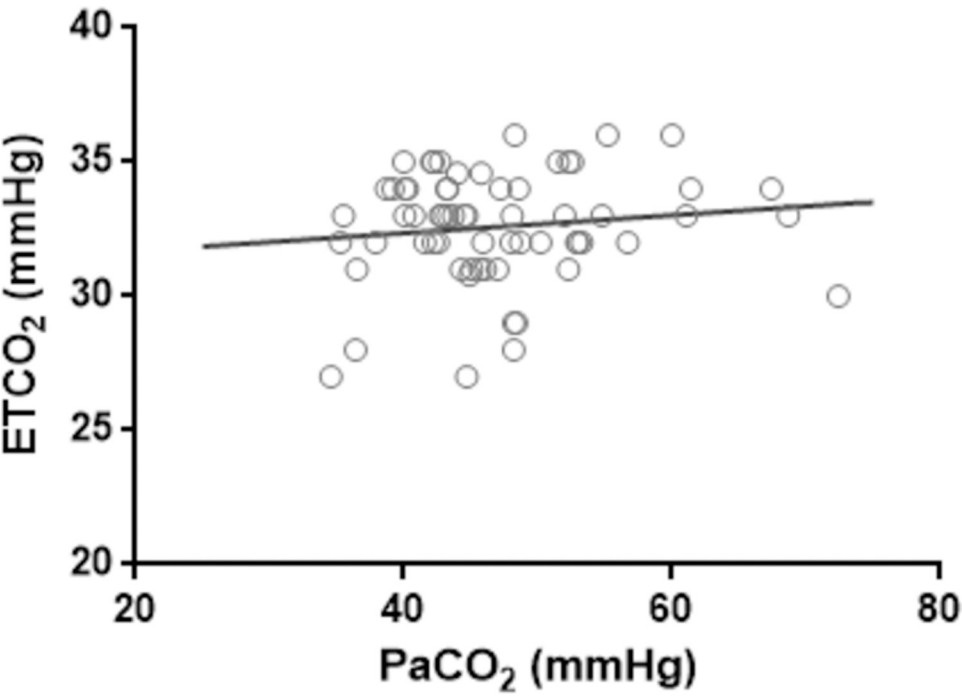

**Fig 6. Comparison between lung function and blood biochemistry.** A Spearman test between two relevant biomarkers of lung function ($ETCO_2$) and blood biochemistry ($PaCO_2$) showed no correlation between these two values. The solid line represents the linear regression. Spearman R-score = 0.046 (p = 0.718).

Based on ultrasound images of lung (S2 Fig in S1 File) and heart (S3 Fig in S1 File), there were no alterations in the functioning of these organs that could indicate injuries related to the use of Masi mechanical ventilator. S4 Table in S1 File shows all of the data recorded for cardio-respiratory parameters. Furthermore, the histopathological analysis performed on the lung, brain, heart, liver and kidney did not show early lesions consistent with barotrauma or hypoxia. However, it was possible to recognize some tissue alterations in these organs, which suggest previous or chronic infection processes (S1B Fig and S1 Table in S1 File).

Swine have anatomical similarities with humans, which make them a good experimental model for respiratory interventions [17, 18]. However, production swine have been selected for larger thoracic cavities resulting in larger tidal volumes with uncertain repercussions on the functional differences from an anatomical and physiological point of view [19, 20]. Medical education in Peru still uses swine as a model for surgical training. Therefore, there is significant experience implementing protocols for sedation and deep anaesthesia similar to those applied by health personnel for intubation process in the Operating Room and ICU [21–23].

At initial conditions ($t_0$), prior to deep anaesthesia and intubation, swine exhibit a good physiological management of gas airway exchange [24]. Despite the fact that most data were located within the rank of previously described standard physiological parameters, the distribution of our baseline data allowed to recognize statistically significant differences. For instance, our swine exhibit high measures of $PaO_2$, as well as increased levels of $HCO_3^-$, BEecf, and $ETCO_2$. This notably changes are accompanied by milder variations (no statistically significant) which includes the decrease of heart rate and the increase of both $PaCO_2$ and respiratory rate. Considering that samples at $t_0$ were taken from sedated animals, hemodynamic alterations regarding the reduction of body inactivity and low corporal temperature might be considered. Thus, these conditions may be responsible for heart rate reduction [25]. Low blood ventilation per minute induces the accumulation of $HCO_3^-$ and $CO_2$, and subsequent rise of $PaCO_2$, BEecf and decrease of $ETCO_2$ parameters. This condition is known as metabolic alkalosis [26, 27]. Increasing of respiratory rate and $PaO_2$ may explain a compensatory measure to improve oxygen availability. It is noteworthy to indicate that once under the effects of deep anaesthesia and connected to the mechanical ventilator, the respiratory rate of the animal is restored and the values returned closer to the average in literature.

Once under the Masi mechanical ventilation system, the control of the respiratory rate and the tidal volume for protective ventilation [12, 28], as well as individualized setting according to swine compliance, allowed as primary outcome the improvement of oxygenation parameters in most patients. For instance, those animals that initially shown $PaO_2$ levels far above the average, reduced their values closer to the standard. Consequently, levels of $PaCO_2$ diminished in short time in contrast to baseline, demonstrating efficient carbon dioxide elimination. These parameters related to oxygen uptake, $PaO_2$ and $PaCO_2$, remained almost constant over time. $PaO_2$ showed slight oscillatory behaviour. Accumulation of $PaCO_2$, usually associated with an ineffective gas exchange process [29], was not observed during our interventions.

Additionally, other parameters to assess the correct oxygen transportation and carbon dioxide elimination during airflow are $SaO_2$ and $ETCO_2$, respectively [15, 16]. According to our results, $SaO_2$ remained at high levels (~ 100%) at all sampling times, including the baseline. It is possible that the body temperature of 37°C in patients during experimental protocol, which is relatively low for the species (normal body temperature = 39°C) but not enough to induce hypothermia (32–34°C) [30, 31], slightly contribute to enhance oxygen affinity to haemoglobin keeping optimum saturation levels [32, 33]. $ETCO_2$ levels were found below the predicted average but its value increases over time, getting closer to the range suggested in the literature. These maximum levels of $ETCO_2$ complemented with the decrease of $PaCO_2$ during the interventions corroborate the correct elimination of carbon dioxide as a waste product of cellular

metabolism [16]. Furthermore, this increase in ETCO2 values is favourable since previous work described that slightly high levels of $ETCO_2$ in humans could be related to decreased odds of lung injury [34, 35]. Therefore, the upper distribution of $ETCO_2$ values is consistent with the absence of lung tissue damage. Similarly, the establishment of uncorrelated behaviour between the $PaCO_2$ and $ETCO_2$ variables represent an additional parameter to suggest no tissue damage [16, 36, 37].

Likewise, variations in $PaCO_2$ directly affect pH and $HCO_3^-$ values. Hypocapnia (low levels of $PaCO_2$) triggered by hyper-ventilatory processes reduces the availability of $CO_2$ molecules for the production of $[HCO_3^-]$ and $[H^+]$. The reduction of these ions leads to an increase in serum pH levels, phenomenon known as acidosis. Conversely, hypercapnia (high levels of $PaCO_2$) leads to lowering serum pH levels, alkalosis [38, 39]. During the use of Masi mechanical ventilator, there was a weakly, but no statistic significant, reduction of $PaCO_2$ at the beginning. This event was reflected immediately in the decrease of $HCO_3^-$ and increase of pH values. Although pH values in humans are around 7.4, normal values in swine are between 7.45–7.55 [10]. Therefore, the second outcome achieved by Masi mechanical ventilator is the proper control of acid-base equilibrium.

Variations on serum lactate concentration may also influence in pH imbalance [38]. Lactate is a common cellular product from anaerobic respiration, in healthy conditions the clearance of lactate is managed by the liver though a process known as gluconeogenesis. Lactate accumulation in blood is usually related to inefficient perfusion, liver failure or tissue damages, and in some cases is considered a predictor of mortality in patients [35, 40]. In this sense, swine exposed to Masi mechanical ventilation exhibited a reduction of serum lactate concentrations over time. Non-accumulation of lactate indicates good perfusion in tissues, and prompts physiological integrity of distal organs. Hence, Masi mechanical ventilator guarantees, as a third outcome, an adequate gas exchange in tissues while maintaining an adequate lactate metabolism and preventing its accumulation. Moreover, in contrast to the gradual decrease of lactate in serum, $HCO_3^-$ levels increased steadily as a compensatory mechanism to keep pH in balance [41]. Furthermore, BEecf measures the acid-base disturbances and have mathematically a direct relationship with $HCO_3^-$ [42], therefore slightly increased values over time mirror $HCO_3^-$ variations.

During the experimental procedures, some patients exhibited outlying physiological parameters from the group mean; these patients were swine 2 and 5. Cases of metabolic acidosis induced by the use of propofol during anaesthesia have been previously described [43]. This metabolic acidosis is characterized by the increase in lactate levels, and the reduction of BEecf and $HCO_3^-$ in blood, which lead to pH decline [43]. Likewise, it has been found that there is a direct relationship between the levels of $HCO_3^-$ and $ETCO_2$ [44]. In accordance to these clinical characteristics, the alterations observed in the hemodynamic parameters of our patients described a mild case of metabolic acidosis. While the causes of this disorder remains unclear, it has been suggested that some subclinical alterations in distal organs could be a risk factor, suggesting that during acidosis stress these alterations may become harder to compensate [43, 45]. According to histopathological analysis, both swine 2 and 5 presented mild kidney disturbance, and severe pulmonary alteration, but neither have early hypoxic lesions. Pig 2 also presented minor liver compromise. Therefore, our observations on arterial blood biochemistry effects are more likely to be connected to increased risk of propofol-mediated metabolic acidosis due to underlying pathologies rather than as a result of the mechanical ventilation.

PSV was largely described as a successful tool to validate the ability to breath autonomously previous to the endotracheal tube removal in patients under mechanical ventilation [46, 47]. PSV capacity to predict a successful extubation procedure (~ 85%), as well as its ability to

maintain stable hemodynamic parameters have been previously described [46, 48]. However, in addition to the device support, some physiological characteristics are related to the probability of a good recovery from assisted ventilation, avoiding reintubation cases [46, 47]. In our attempt to reduce unnecessary animal suffering, we monitor the autonomic respiratory capacity in sedated animals. In this way, we observed that after the use of Masi mechanical ventilator, all the animals were able to recover their spontaneous breathing.

Finally, the physiological changes produced during the use of the mechanical ventilator cause stress at the lung, but can also affect distant organs. Therefore, mechanical ventilators could produce functional alterations and even injuries in heart, liver, kidney and brain [49, 50]. For this reason, and as a final outcome, the post-mortem analysis certified the absence of both acute tissue damage and barotrauma signals due to the use of Masi. Thus, pathology studies corroborate what was expected according to the previously described biological indicators, such as slightly elevated levels of $ETCO_2$ and the non-correlation between $ETCO_2$ and $PaCO_2$.

## Conclusions

In this study, we demonstrate that a mechanical ventilator that operates by controlling an automated manual resuscitator or self-inflating bag like Masi preserves patient physiological parameters within normal ranges during acute exposure. The post-mortem study of critical organs and histopathological observations present no evident signs of barotrauma caused concluding that the Masi mechanical ventilator is safe to use in the pre-clinical trial. Our data shows successful control of blood biochemistry mechanisms involved with oxygenation supply and carbon dioxide removal that include: Oxygen uptake ($PaO_2$, $PaCO_2$), carbon dioxide release ($ETCO_2$), oxygen transport ($SaO_2$), tissue perfusion (lactate), acid-base balance ($HCO_3^-$, pH). Therefore, Masi performs body gas exchange in a similar way as other commercial mechanical ventilators.

## Supporting information

**S1 File.**
(PDF)

## Acknowledgments

The authors would like to thank to all the members of the Masi design team, especially to all of the collaborators working at the five institutions involved in this project (BREIN, DIACSA, EAT, and Zolid Design). Without all of their effort, professionalism and sacrifice while working steadily during the pandemic, this device would have not existed. We are thankful to Gisela Fernandez-Rivas Plata, Jordi Lopez-Tremoleda and Ricardo Hora for their insightful comments on study design. Also, the authors are thankful to all of the private in-kind donations that funded manufacturing the ventilators that are named at https://www.proyectomasi.pe/

## Author Contributions

**Conceptualization:** Cesar Miguel Gavidia, Roberto Davila Fernandez, Juan Alberto Vargas Zuñiga, Alberto Crespo Paiva, Juan Fernando Calcina Isique, Jaime Reategui, Benjamin Castañeda, Fanny L. Casado.

**Data curation:** Maryanne Melanie Gonzales Carazas, Fanny L. Casado.

**Formal analysis:** Maryanne Melanie Gonzales Carazas, Rosa Perales, Fanny L. Casado.

**Funding acquisition:** Fanny L. Casado.

**Investigation:** Cesar Miguel Gavidia, Roberto Davila Fernandez, Juan Alberto Vargas Zuñiga, Alberto Crespo Paiva, William Bocanegra, Joan Calderon, Evelyn Sanchez, Rosa Perales, Brandon Zeña, Juan Fernando Calcina Isique, Jaime Reategui, Benjamin Castañeda, Fanny L. Casado.

**Methodology:** Juan Alberto Vargas Zuñiga, Alberto Crespo Paiva, William Bocanegra, Joan Calderon, Evelyn Sanchez, Rosa Perales, Brandon Zeña, Juan Fernando Calcina Isique, Jaime Reategui, Fanny L. Casado.

**Project administration:** Cesar Miguel Gavidia, Juan Fernando Calcina Isique, Fanny L. Casado.

**Resources:** Cesar Miguel Gavidia, Roberto Davila Fernandez, Juan Alberto Vargas Zuñiga, Alberto Crespo Paiva, William Bocanegra, Rosa Perales, Juan Fernando Calcina Isique, Jaime Reategui, Benjamin Castañeda.

**Supervision:** Cesar Miguel Gavidia, Roberto Davila Fernandez, Fanny L. Casado.

**Visualization:** Juan Alberto Vargas Zuñiga, Rosa Perales.

**Writing – original draft:** Maryanne Melanie Gonzales Carazas.

**Writing – review & editing:** Cesar Miguel Gavidia, Roberto Davila Fernandez, Juan Alberto Vargas Zuñiga, Alberto Crespo Paiva, William Bocanegra, Joan Calderon, Evelyn Sanchez, Rosa Perales, Brandon Zeña, Juan Fernando Calcina Isique, Jaime Reategui, Benjamin Castañeda, Fanny L. Casado.

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
