## [Decision Letter · Decision Letter 0]

23 Nov 2021

PONE-D-21-30195Biological evaluation of a mechanical ventilator that operates by controlling an automated manual resuscitator. A descriptive study in swinePLOS ONE

Dear Dr. Fanny Lys Casado,

Thank you for submitting your manuscript to PLOS ONE. After careful consideration, we feel that it has merit but does not fully meet PLOS ONE’s publication criteria as it currently stands. Therefore, we invite you to submit a revised version of the manuscript that addresses the points raised during the review process.

ACADEMIC EDITOR: Thank you very much for having submitted this paper for consideration. This paper is of potential interest however it needs to be revised according to the reviewers' comments. 

We look forward to receiving your revised manuscript.

Kind regards,

Simone Savastano

Academic Editor

PLOS ONE

Journal Requirements:

Additional Editor Comments:

Thank you very much for having submitted this paper for consideration. This paper is of potential interest however it needs to be revised according to the reviewers' comments.

Reviewers' comments:

Reviewer's Responses to Questions

**Comments to the Author**

1. Is the manuscript technically sound, and do the data support the conclusions?

Reviewer #1: Yes

Reviewer #2: Partly

2. Has the statistical analysis been performed appropriately and rigorously? 

Reviewer #1: Yes

Reviewer #2: No

3. Have the authors made all data underlying the findings in their manuscript fully available?

Reviewer #1: Yes

Reviewer #2: Yes

4. Is the manuscript presented in an intelligible fashion and written in standard English?

Reviewer #1: Yes

Reviewer #2: No

5. Review Comments to the Author

Reviewer #1: Dear editor

Dear Authors

This is a descriptive preclinical study describing the physiological effect of a portable, low-cost mechanical ventilator (Masi) on a swine model; since the demand of ventilators increased dramatically worldwide during the recent pandemic, the potential of this device in terms of ease of production is of the utmost interest. The authors investigate the effects of six hours of mechanical ventilation on eight healthy pigs in terms of variation of biological parameters:

• Gas exchange and pulmonary function

• Acid-base disturbances

• Lactate and HCO3- concentration

• Cardiac output and heart rate

• Hemoglobin

The protocol then included histological analysis of lung, kidney, liver, brain and heart of the subjects, failing to demonstrate any major organ damage and in particular any hypoxic lesions relatable to the mechanical ventilation.The researchers concluded that Masi preserves patient physiological parameters within normal ranges during acute exposure (6 hours), avoiding barotrauma.

The study has several advantages, such as the reliable animal model (swine) and the absence of evident macroscopic and microscopic organ damage secondary to mechanical ventilation with Masi, as well as the absence of any major organ dysfunction. It is interesting to note that the values of pulmonary compliance of the pigs as listed by the authors in TABLE 2 (between 37 and 44 ml/cmH20) is inferior to the normal human values of 54 [44-64] ml/cmH20 but similar to the value reported in human patients suffering from ARDS - 39 [32-50] mL/cm H2O (1), which are the patients who one would expect would need most often a mechanical ventilator. Considering this, the safety in terms of barotrauma showed by the ventilator is of even greater significance. At the end of the study period every animal was able, when ventilation was stopped, to perform spontaneous breathing supported by PSV, a strong indicator of the ability to breath autonomously previous to the endotracheal tube removal in patients under mechanical ventilation.

There are however some issues:

Major Issue

• The first and most important limitation of the study is the short term of the follow up during mechanical ventilation (six hours). Since the aim of this animal study is to validate the ventilator, one could argue that the duration of mechanical ventilation in ARDS or Sars-CoV-2 pneumonia with respiratory failure is seldom limited to a few hours, extending well beyond such as several days in many reports (2) .

Minor Issues

• Even though I understand that is not strictly the point of this article, a simple diagram of the mechanical ventilator itself (such as figure 1 of the article in the fifth citation, which I understand is also from your group) could help the reader to understand the item in discussion.

• Linear measurements in short axis (as shown in figure S2 of the supporting information) are not the echocardiographic assessment of choice for cardiac output. It would be reasonable in terms of accuracy to present the data in terms of LVEDV, LVESV and HR instead of cardiac output (being the clinical value most likely the same).

• Line 405 “Therefore, the upper distribution of ETCO2 values is consistent the absence of lung tissue damage” should be “is consistent with the absence of lung tissue damage”

(1) Arnal JM, Garnero A, Saoli M, Chatburn RL. Parameters for Simulation of Adult Subjects During Mechanical Ventilation. Respir Care. 2018 Feb;63(2):158-168. doi: 10.4187/respcare.05775. Epub 2017 Oct 17. PMID: 29042486.

(2) King CS, Sahjwani D, Brown AW, et al. Outcomes of mechanically ventilated patients with COVID-19 associated respiratory failure. PLoS One. 2020;15(11):e0242651. Published 2020 Nov 23. doi:10.1371/journal.pone.0242651

Nice work!

Reviewer #2: The manuscript entitled “Biological evaluation of a mechanical ventilator that operates by controlling an automated manual resuscitator. A descriptive study in swine” ( Manuscript number PONE-D-21-30195) presented the characteristics of a new ventilator device.

I congratulate the authors for the study, risen from the need to dispose of larger amount of mechanical ventilation during COVID- 19 pandemics; this reason makes this study relevant.

However, there are several points that needs to be addressed in order to make this paper worthy of publication.

The characteristic of the tested ventilator highlighted in the title (“…that operates by controlling an automated manual resuscitator”) is not considered at all into the manuscript.

Introduction. In my opinion, retracing the evolution of the COVID- 19 pandemics (i.e. Specifying date of the first case in China and in Perù) is not pertaining to the aim of the study. In order to put the new mechanical ventilator production into the context of the COVID – 19 pandemics, the authors should instead provide more precise epidemiologic data (es. number of patients admitted, patients requiring mechanical ventilation).

From line 76 to 79. Please provide some details about the so called “operating mechanism” of the Masi ventilator. In this context, mentioning the commercial name of other similar ventilator may be confounding and not relevant. Which is the meaning of “less complex mechanical ventilator systems”? In other words, please brefly describe the differences between the Masi ventilator and others, commonly used in the ICU.

From line 85 to 87. Please rephrase this sentence.

From line 96 to 98.The aims of the study should be precisely defined at this point, together with the primary endpoints.

Methods. The experimental design should be clearer then it is. Moreover, sample size calculation should be explicated for each proposed study included in the experimental design.

Line 216-217. Variables measured by echocardiography should be defined and subsequently described into the results section. The assessment “there were no alterations in the functioning of these organs that could indicate injuries related to the use of Masi mechanical ventilator” (line348-349) is a qualitative evaluation.

Line 251. Pigs (at baseline) are not patients.

Table 3. Reference values showed in a separated figure are unclear.

6. PLOS authors have the option to publish the peer review history of their article (what does this mean?). If published, this will include your full peer review and any attached files.

Reviewer #1: **Yes: **Alessandro Fasolino

Reviewer #2: No

---

## [Author Response · Author response to Decision Letter 0]

20 Jan 2022

ANSWER The Manuscript has been modified into PLOS One format.

ANSWER We have included all of the raw data in the Supporting Information file.

ANSWER The Manuscript has been modified to address this requirement.

ANSWER We modified the Supporting Information files and have updated the captions and in-text citations to address this requirement.

II. COMMENTS OF REVIEWER #1: 

This is a descriptive preclinical study describing the physiological effect of a portable, low-cost mechanical ventilator (Masi) on a swine model; since the demand of ventilators increased dramatically worldwide during the recent pandemic, the potential of this device in terms of ease of production is of the utmost interest. The authors investigate the effects of six hours of mechanical ventilation on eight healthy pigs in terms of variation of biological parameters:

• Gas exchange and pulmonary function

• Acid-base disturbances

• Lactate and HCO3- concentration

• Cardiac output and heart rate

• Hemoglobin

The protocol then included histological analysis of lung, kidney, liver, brain and heart of the subjects, failing to demonstrate any major organ damage and in particular any hypoxic lesions relatable to the mechanical ventilation.The researchers concluded that Masi preserves patient physiological parameters within normal ranges during acute exposure (6 hours), avoiding barotrauma.

The study has several advantages, such as the reliable animal model (swine) and the absence of evident macroscopic and microscopic organ damage secondary to mechanical ventilation with Masi, as well as the absence of any major organ dysfunction. It is interesting to note that the values of pulmonary compliance of the pigs as listed by the authors in TABLE 2 (between 37 and 44 ml/cmH20) is inferior to the normal human values of 54 [44-64] ml/cmH20 but similar to the value reported in human patients suffering from ARDS - 39 [32-50] mL/cm H2O (1), which are the patients who one would expect would need most often a mechanical ventilator. Considering this, the safety in terms of barotrauma showed by the ventilator is of even greater significance. At the end of the study period every animal was able, when ventilation was stopped, to perform spontaneous breathing supported by PSV, a strong indicator of the ability to breath autonomously previous to the endotracheal tube removal in patients under mechanical ventilation.

(1) Arnal JM, Garnero A, Saoli M, Chatburn RL. Parameters for Simulation of Adult Subjects During Mechanical Ventilation. Respir Care. 2018 Feb;63(2):158-168. doi: 10.4187/respcare.05775. Epub 2017 Oct 17. PMID: 29042486.

There are however some issues:

Major Issue

• The first and most important limitation of the study is the short term of the follow up during mechanical ventilation (six hours). Since the aim of this animal study is to validate the ventilator, one could argue that the duration of mechanical ventilation in ARDS or Sars-CoV-2 pneumonia with respiratory failure is seldom limited to a few hours, extending well beyond such as several days in many reports (2).

(2) King CS, Sahjwani D, Brown AW, et al. Outcomes of mechanically ventilated patients with COVID-19 associated respiratory failure. PLoS One. 2020;15(11):e0242651. Published 2020 Nov 23. doi:10.1371/journal.pone.0242651

ANSWER: We agree with the reviewer that the aim of the study is to validate the ventilator and that there are clinical scenarios that we do not address in this study. However, it needs to be qualified that this is a pre-clinical validation of a medical device. Consensual technical standards such as the ISO10993-2 Biological Evaluation of Medical Devices – Part 2: Animal Welfare Requirements define validation as the “formal process by which the reliability and relevance of a test method is established for a particular purpose”. In that sense, testing the device for six-hours in pigs is a reliable and relevant method to address questions of control of gas exchange and acute damage to internal organs. The purpose of pre-clinical validation is to provide evidence to support moving into clinical validation where questions regarding time of exposure in the context of specific diseases are currently being addressed. Therefore, we respectfully disagree that a six-hour exposure represents any limitation to this specific study since it was not designed to address how long it takes to observe detrimental effects in humans diagnosed with a particular disease because those questions are better suited to be asked in clinical studies with the proper criteria of inclusion and exclusion of participants in the context of pharmaceutical interventions to stabilize the patients or cure the disease. 

Minor Issues

• Even though I understand that is not strictly the point of this article, a simple diagram of the mechanical ventilator itself (such as figure 1 of the article in the fifth citation, which I understand is also from your group) could help the reader to understand the item in discussion.

ANSWER: We added a new figure as Figure 1 and modified the introduction to better explain the characteristics of the ventilator being studied.

• Linear measurements in short axis (as shown in figure S2 of the supporting information) are not the echocardiographic assessment of choice for cardiac output. It would be reasonable in terms of accuracy to present the data in terms of LVEDV, LVESV and HR instead of cardiac output (being the clinical value most likely the same).

ANSWER: We have modified S2 Figure caption to better communicate its relevance and modified S4 Table to provide all of the data recorded including LVEDV, LVESV, HR and cardiac output.

• Line 405 “Therefore, the upper distribution of ETCO2 values is consistent the absence of lung tissue damage” should be “is consistent with the absence of lung tissue damage” Nice work!

ANSWER: Thank you for your encouraging words. We fixed this involuntary typographical error on the Manuscript. 

III. COMMENTS OF REVIEWER #2: 

The manuscript entitled “Biological evaluation of a mechanical ventilator that operates by controlling an automated manual resuscitator. A descriptive study in swine” ( Manuscript number PONE-D-21-30195) presented the characteristics of a new ventilator device.

I congratulate the authors for the study, risen from the need to dispose of larger amount of mechanical ventilation during COVID- 19 pandemics; this reason makes this study relevant.

However, there are several points that needs to be addressed in order to make this paper worthy of publication.

The characteristic of the tested ventilator highlighted in the title (“…that operates by controlling an automated manual resuscitator”) is not considered at all into the manuscript.

ANSWER: We added a new figure as Figure 1 and modified the introduction to better explain the characteristics of the ventilator being studied.

Introduction. In my opinion, retracing the evolution of the COVID- 19 pandemics (i.e. Specifying date of the first case in China and in Perù) is not pertaining to the aim of the study. In order to put the new mechanical ventilator production into the context of the COVID – 19 pandemics, the authors should instead provide more precise epidemiologic data (es. number of patients admitted, patients requiring mechanical ventilation).

ANSWER: We modified the introduction to include the information requested.

From line 76 to 79. Please provide some details about the so called “operating mechanism” of the Masi ventilator. In this context, mentioning the commercial name of other similar ventilator may be confounding and not relevant. Which is the meaning of “less complex mechanical ventilator systems”? In other words, please brefly describe the differences between the Masi ventilator and others, commonly used in the ICU.

ANSWER: We added a new figure as Figure 1 and modified the introduction to better explain the characteristics of the ventilator being studied in the context of other ventilators found in ICU.

From line 85 to 87. Please rephrase this sentence.

ANSWER: We modified the introduction to address this comment.

From line 96 to 98.The aims of the study should be precisely defined at this point, together with the primary endpoints.

ANSWER: We modified the introduction to include the information requested.

Methods. The experimental design should be clearer then it is. Moreover, sample size calculation should be explicated for each proposed study included in the experimental design.

ANSWER: We modified original Figure 1 to be new Figure 2 and modified the text of Experimental design to better explain the characteristics of the ventilator being studied.

Line 216-217. Variables measured by echocardiography should be defined and subsequently described into the results section. The assessment “there were no alterations in the functioning of these organs that could indicate injuries related to the use of Masi mechanical ventilator” (line348-349) is a qualitative evaluation.

ANSWER: We have modified S2 Figure caption to better communicate its relevance and modified S4 Table to provide all of the data recorded that supports the qualitative evaluation provided by the veterinary cardiologist.

Line 251. Pigs (at baseline) are not patients.

ANSWER: We modified the text to fix this involuntary error.

Table 3. Reference values showed in a separated figure are unclear.

ANSWER: For clarity, Table 3 was modified to only focus on summarized data from current study. We added S2 Table to showcase the comparison between swine physiological data from Table 1 (published previously) and Table 3 (current pre-clinical study).

---

## [Decision Letter · Decision Letter 1]

17 Feb 2022

Biological evaluation of a mechanical ventilator that operates by controlling an automated manual resuscitator. A descriptive study in swine

PONE-D-21-30195R1

Dear Dr. Casado,

We’re pleased to inform you that your manuscript has been judged scientifically suitable for publication and will be formally accepted for publication once it meets all outstanding technical requirements.

Kind regards,

Simone Savastano

Academic Editor

PLOS ONE

Additional Editor Comments (optional):

I congratulate the Authors for having addressed properly the Reviewers' comments. There are still some typos that can be fixed in the further editing phases following the reviewers' indications.

Reviewers' comments:

Reviewer's Responses to Questions

**Comments to the Author**

1. If the authors have adequately addressed your comments raised in a previous round of review and you feel that this manuscript is now acceptable for publication, you may indicate that here to bypass the “Comments to the Author” section, enter your conflict of interest statement in the “Confidential to Editor” section, and submit your "Accept" recommendation.

Reviewer #1: All comments have been addressed

Reviewer #2: All comments have been addressed

2. Is the manuscript technically sound, and do the data support the conclusions?

Reviewer #1: Yes

Reviewer #2: Yes

3. Has the statistical analysis been performed appropriately and rigorously? 

Reviewer #1: Yes

Reviewer #2: Yes

4. Have the authors made all data underlying the findings in their manuscript fully available?

Reviewer #1: Yes

Reviewer #2: Yes

5. Is the manuscript presented in an intelligible fashion and written in standard English?

Reviewer #1: Yes

Reviewer #2: Yes

6. Review Comments to the Author

Reviewer #1: All my concerns were addressed by the authors.

Minor issues/typos:

- I don't understand the correction in the following sentence "Atypical responses in Pig #1_2 and Pig #1_5" in line 323. Maybe it should be: "Atypical responses in Pig 2 and Pig 5"?

- "Figure 54" in line 387/402/404 is probably a typo

Once addressed these very minor issues the article is fit for publication.

Reviewer #2: The authors have clarified several of the questions I raised in my previous review. The paper is worthy of publication. The authors only should revise the language to further improve readability.

7. PLOS authors have the option to publish the peer review history of their article (what does this mean?). If published, this will include your full peer review and any attached files.

Reviewer #1: No

Reviewer #2: No

---

## [Editor Report · Acceptance letter]

23 Feb 2022

PONE-D-21-30195R1 

Biological evaluation of a mechanical ventilator that operates by controlling an automated manual resuscitator. A descriptive study in swine 

Dear Dr. Casado:

I'm pleased to inform you that your manuscript has been deemed suitable for publication in PLOS ONE. Congratulations! Your manuscript is now with our production department. 

Kind regards, 

on behalf of

Dr. Simone Savastano 

Academic Editor

PLOS ONE